# Gender Differences in Clinical and Biochemical Variables of Patients Affected by Bipolar Disorder

**DOI:** 10.3390/brainsci15020214

**Published:** 2025-02-19

**Authors:** Luigi Piccirilli, Enrico Capuzzi, Francesca Legnani, Martina Di Paolo, Anna Pan, Alessandro Ceresa, Cecilia Maria Esposito, Luisa Cirella, Teresa Surace, Ilaria Tagliabue, Massimo Clerici, Massimiliano Buoli

**Affiliations:** 1Department of Neurosciences and Mental Health, Fondazione IRCCS Ca’ Granda Ospedale Maggiore Policlinico, 20122 Milan, Italy; luigi.piccirilli@unimi.it (L.P.); francesca.legnani@unimi.it (F.L.); martina.dipaolo@unimi.it (M.D.P.); anna.pan@unimi.it (A.P.); cecilia.esposito@policlinico.mi.it (C.M.E.); 2Department of Mental Health, Fondazione IRCCS San Gerardo dei Tintori, via G.B. Pergolesi 33, 20900 Monza, Italy; e.capuzzi1@campus.unimib.it (E.C.); teresa.surace@irccs-sangerardo.it (T.S.); ilaria.tagliabue@ircce-sangerardo.it (I.T.); 3Healthcare Professionals Department, Foundation IRCCS Ca’ Granda Ospedale Maggiore Policlinico, 20122 Milan, Italy; luisa.cirella@policlinico.mi.it; 4Department of Medicine and Surgery, University of Milano Bicocca, 20900 Monza, Italy; massimo.clerici@unimib.it; 5Department of Pathophysiology and Transplantation, University of Milan, 20122 Milan, Italy

**Keywords:** bipolar disorder (BD), gender differences, clinical variables, biochemical parameters

## Abstract

**Introduction:** Bipolar disorder (BD) affects over 1% of the global population and significantly impacts psychosocial functioning and life expectancy. This manuscript has the objective of investigating gender differences in the clinical and biochemical parameters of patients affected by BD. **Methods:** This retrospective cross-sectional study examined 672 patients diagnosed with BD in psychiatric wards in Milan and Monza. Clinical data and biochemical parameters were collected on the first day of hospitalization. Independent sample *t*-tests, chi-square tests and binary logistic regressions were performed to identify gender differences in BD. **Results:** With regard to univariate analyses, women were found to be more susceptible to psychiatric comorbidities (χ^2^ = 12.75, *p* < 0.01), medical comorbidities (χ^2^ = 45.38, *p* < 0.01), obesity (χ^2^ = 6.75, *p* = 0.01) and hypercholesterolemia (χ^2^ = 23.54, *p* < 0.01), as well as to having more mood episodes in the year prior to hospitalization (t = 5.69, *p* < 0.01). Men were found to be more likely to develop psychotic symptoms (χ^2^ = 4.40, *p* = 0.04), to be tobacco smokers (χ^2^ = 15.13, *p* < 0.01) and to have substance abuse disorders (χ^2^ = 14.66, *p* = <0.01). Logistic regression analyses showed that women compared to men showed more psychiatric comorbidity (*p* < 0.01), higher Global Assessment of Functioning (GAF) scores (*p* = 0.05) and higher total cholesterol plasma levels (*p* < 0.01); however, they also had fewer red blood cells (*p* < 0.01) and lower creatinine plasma levels (*p* < 0.01). **Conclusions:** Female patients (compared to males) exhibited higher levels of global functioning despite the higher frequency of psychiatric comorbidity and susceptibility to metabolic complications; consistent with earlier studies, female patients also showed higher cholesterol levels. Further studies will have to confirm the present findings and identify gender-related clinical pathways for the management of BD.

## 1. Introduction

Bipolar disorder (BD) is a severe and chronic condition that affects more than 1% of the global population [1]. This illness significantly impairs psychosocial functioning and can reduce life expectancy by 10–20 years [2,3].

A large number of clinical factors were studied in relation to the course of BD. Of note, an earlier age of onset was associated with numerous negative long-term outcomes, such as a higher probability of psychotic symptoms, more suicide attempts and fewer opportunities for psychological intervention [4,5]. Furthermore, depressive polarity at onset seems to increase the risk of suicidal behaviour, whereas a manic episode at onset is associated with delusions and raises vulnerability in regard to future manic episodes [5]. The duration of untreated illness (DUI) represents another variable related to unfavourable outcomes, such as a higher frequency and number of suicide attempts, as well as more depressive episodes and medical comorbidities [6,7]. Another clinical aspect that affects BD prognosis is the presence of rapid cycling that increases the risk of hospitalizations and suicidal behaviours [8,9].

The mentioned unfavourable clinical factors can reflect the severity of biochemical abnormalities associated with BD [10]. Of note, several studies identified over-inflammation in subjects affected by BD in comparison with healthy controls [11]. Neutrophil-to-lymphocyte ratio (NLR) and monocyte-to-lymphocyte ratio (MLR) were found to be more increased in bipolar patients than in healthy controls [12], especially during manic [12] or hypomanic phases [13]. Similarly, the plasma concentrations of uric acid seem to be higher in bipolar patients than in healthy controls and subjects affected by schizophrenia or major depressive disorders [14]. Lower cholesterol levels were detected in bipolar men with suicide attempts versus men without suicide attempts [15], while thyroid dysfunction is possibly associated with rapid cycling [16].

All the mentioned factors affecting the course of BD can be different in males versus females. Bipolar women seem to be more prone to developing rapid cycling [17] and medical comorbidities (e.g., thyroid dysfunction) [18], while men with BD suffer more frequently from substance use disorders than women [19]. With regard to biological parameters, women affected by hypomania were found to have higher platelets than men; conversely, male hypomanic patients showed higher NLR values than female ones [20]. Furthermore, gender differences in the plasma levels of malondialdehyde, a marker of lipid peroxidation, were reported in bipolar patients [21]. Finally, c-reactive protein (CRP) levels were found to be a predictor of cognitive performance in bipolar women but not in men [22].

In the framework of precision psychiatry, the purpose of the present manuscript is to investigate gender differences in clinical and biochemical parameters of subjects affected by BD. Despite the presence of preliminary findings, more data are probably needed to clarify this topic.

## 2. Methods

This is a retrospective cross-sectional study. A sample of 672 patients (277 males and 395 females) who have been admitted to the psychiatric wards of the hospitals of Milan (N = 522) and Monza (N = 150) and diagnosed as being affected by BD, according to DSM criteria, was included. The patients were hospitalized from 2008 to 2023 so that DSM-IV-TR (text revision) or DSM-5 criteria were applied. In cases of multiple hospitalizations, the most recent hospitalization was taken into account.

Patients with the following criteria were included: (1) age ≥ 18 years old and (2) diagnosis of BD. If the subjects satisfied the criteria for other psychiatric disorders, BD represented the main diagnosis (associated with more social dysfunction). The information of the presence of psychiatric comorbidity was collected. Exclusion criteria were represented by (1) age < 18 years, (2) current pharmacological treatment associated with mood alterations (steroids, levetiracetam, interferon, efavirenz) [23], (3) re-exacerbation of medical comorbidities that could affect significantly biochemical parameters or justify mood dysregulation (e.g., rheumatoid arthritis) [24], (4) peripartum (pregnancy and one month after delivery), as this period is characterized by susceptibility to mood disorders and biological changes affecting biochemical parameters [25]. Clinical and biochemical data were obtained by clinical charts (both electronic ones and SIPRL Lombardy electronic databases) and intranet hospital applications. If any information was lacking, this was requested from patients or their relatives/caregivers. Blood analyses were performed the first day of admission, and patients, once discharged, were followed up with by community health services. The protocol was approved by the local Ethical Committee (Fondazione IRCCS Ca’Granda Ospedale Maggiore Policlinico) (approval number 1789, 9 February 2022).

The following variables were collected:

Clinical variables included age at admission, gender, age at illness onset (this was determined based on previous clinical documentation and the medical history collected from patients themselves and, when impossible, their family member or caregivers), duration of hospitalization, duration of untreated illness (DUI), duration of illness, number of previous psychiatric hospitalizations, number of previous acute episodes, number of previous manic episodes, number of previous hypomanic episodes, number of previous depressive episodes, number of previous suicide attempts, number of manic episodes in the last year, number of hypomanic episodes in the last year, number of depressive episodes in the last year, cumulative number of mood episodes in the last year, number of previous substance-induced episodes, current Young Mania Rating Scale (YMRS) score [26], current Hamilton Depression Rating Scale 21 items (HAM-D) score [27], current Montgomery–Asberg Depression Rating Scale (MADRS) score [28], current Brief Psychiatric Rating Scale (BPRS) score for general psychopathology [29], current Hamilton Anxiety Rating Scale (HAM-A) score [30], current Global Assessment of Functioning (GAF) score [31], Body Mass Index (BMI), type of current episode, current presence of mixed features, lifetime presence of mixed features, lifetime presence of psychotic symptoms, type of bipolar disorder, lifetime presence of rapid cycling, lifetime presence of seasonality, family history of psychiatric disorders, multiple family history of psychiatric disorders, presence of lifetime substance use disorders, presence of lifetime alcohol use disorders, presence of lifetime multiple substance use disorders, current tobacco smoking, cigarettes/day, type of the last mood episode, administration of poly-therapy during the last mood episode (intended as any combination of more than a single psychotropic drug), comorbid personality disorder (DSM-IV-TR or DSM-5 criteria), psychiatric comorbidity, previous suicide attempts, medical comorbidity, medical poly-comorbidity, obstetrical complications, comorbidity with thyroid diseases, comorbidity with diabetes, comorbidity with hypercholesterolemia, comorbidity with obesity (defined as a BMI ≥ 30) [32], achievement of treatment response in the current episode, achievement of remission in the current episode, current treatment with statins and current treatment with levothyroxine.

Biomarker values included sodium, potassium, white blood cells, red blood cells, hemoglobin, mean corpuscular volume (MCV), platelets, mean platelet volume (MPV), neutrophils, lymphocytes, blood glucose, urea, creatinine, uric acid, aspartate transaminase (AST), alanine transaminase (ALT), gamma-glutamyl transferase (GGT), bilirubin, plasmatic protein, albumin, lactate dehydrogenase (LDH), creatine phosphokinase (CPK), pseudocholinesterase (PCHE), total cholesterol, high-density lipoproteins (HDL), low-density lipoproteins (LDL), triglycerides, blood iron, thyroid-stimulating hormone (TSH), c-reactive protein (PCR), neutrophil to lymphocyte ratio (NLR), platelet to lymphocyte ratio (PLR), AST/ALT ratio and sodium/potassium ratio.

The DUI was defined as the time between the onset of illness and the beginning of a proper pharmacological treatment (mood stabilizers or atypical antipsychotics) [6]. Treatment response was defined by a reduction of at least 50% of the baseline total rating scale scores with the exception of GAF [33]. Remission in the current episode was defined as an endpoint HAM-D score < 8 and a YMRS score < 10 [33].

Statistical analyses were performed through The Statistical Package for Social Sciences (SPSS) for Windows (version 28.0). Descriptive analyses of the total sample were initially performed, and then groups identified according to gender were compared using Student’s *t* tests for continuous variables and χ^2^ tests for qualitative variables (with an odds ratio (OR) calculation and a 95% confidence interval (CI) where appropriate). Finally, statistically significant variables from these univariate analyses were inserted in binary logistic regression models as predictors, with gender acting as dependent variable. Three intermediate models were performed: one with clinical qualitative variables, one with continuous clinical variables and one with biochemical parameters. The variables that were found to be statistically significant in these intermediate models were then inserted in a new final model. The goodness of the model was assessed by the Omnibus and Hosmer–Lemeshow tests.

The level of statistical significance was set at *p* ≤ 0.05.

## 3. Results

The total sample included 672 patients; 41.2% (N = 277) were males and 58.8% (N = 395) were females. None of the subjects described themselves as transgender. Descriptive analyses and a summary of the results of univariate analyses are reported in Table 1 (qualitative variables) and Table 2 (continuous variables).

Information about the pharmacological treatment at the beginning of hospitalization was available for 259 patients: 4 did not engage in pharmacotherapy, 62 were treated with valproic acid, 64 with lithium, 3 with risperidone, 16 with haloperidol, 4 with paliperidone, 23 with olanzapine, 24 with quetiapine, 10 with aripiprazole, 3 with lamotrigine, 1 with asenapine, 9 with Selective Serotonin Reuptake Inhibitors (SSRIs), 11 with Selective Serotonin Noradrenaline Reuptake Inhibitors (SNRIs), 4 with zuclopenthixol, 3 with gabapentin, 3 with carbamazepine, 4 with vortioxetine, 2 with mirtazapine, 2 with trazodone, 2 with tricyclic antidepressants, 1 with lurasidone, 1 with olanzapine pamoate, 2 with paliperidone palmitate and 1 with aripiprazole depot. No differences between genders were detected in relation to treatment (χ^2^ = 20.53, *p* = 0.714), even in relation to the single compounds (*p* < 0.05).

A comparison of female versus male bipolar patients presented the following results:(a)In relation to clinical variables: women resulted to suffer less frequently from psychotic symptoms during life (χ^2^ = 4.40, *p* = 0.04), to be less susceptible to substance use disorders in the lifespan (χ^2^ = 14.66, *p* < 0.01), to suffer more frequently from a comorbid personality disorder (χ^2^ = 6.88, *p* < 0.01) and another psychiatric comorbidity (χ^2^ = 12.75, *p* < 0.01), to be more frequently affected by medical comorbidity (χ^2^ = 45.38, *p* < 0.01) and poly-comorbidity (χ^2^ = 39.06, *p* < 0.01), including thyroid diseases (χ^2^ = 36.60, *p* < 0.01), hypercholesterolemia (χ^2^ = 23.54, *p* < 0.01) and obesity (χ^2^ = 6.75, *p* = 0.01). In addition, female patients (compared to men) were older at admission (t = 2.93, *p* < 0.01), had a later age at onset (t = 2.13, *p* = 0.03), smoked fewer cigarettes (t = 3.96, *p* < 0.01) and were less frequently smokers (χ^2^ = 15.13, *p* < 0.01), less frequently achieved remission in the current episode (χ^2^ = 6.90, *p* = 0.01), showed a higher number of previous depressive episodes (t = 1.99, *p* = 0.04), more previous suicide attempts (t = 1.94, *p* = 0.05), more manic (t = 3.82, *p* < 0.01) and hypomanic episodes (t = 1.98, *p* = 0.05) and more total mood episodes in the last year (t = 5.69, *p* < 0.01).(b)In relation to severity of disorder at hospital admission (rating scale scores): At the moment of admission, women (compared to men) were found to have a more severe psychopathology, as shown by higher scores on the YMRS (t = 2.36, *p* = 0.02), HAM-D (t = 2.25, *p* = 0.03) and BPRS (t = 2.70, *p* = < 0.01), but better functioning, as demonstrated by higher GAF scores (t = 2.34, *p* = 0.02).(c)With regard to biochemical parameters, female bipolar patients than male counterpart had fewer white blood cells (t = 2.81, *p* = 0.01) and neutrophils (t = 2.73, *p* = 0.01), fewer red blood cells (t = 11.81, *p* < 0.01) and hemoglobin (t = 14.34, *p* < 0.01) and a larger MCV (t = 2.32, *p* < 0.02). In addition, creatinine concentration was lower in women compared to men (t = 4.88, *p* < 0.01) similarly to AST (t = 2.22, *p* = 0.03), ALT (t = 4.89, *p* < 0.01), bilirubin (t = 4.24, *p* < 0.01), plasmatic protein (t = 2.07, *p* = 0.04), albumin (t = 3.51, *p* < 0.01), CPK (t = 3.33, *p* < 0.01). On the other hand, both total cholesterol (t = 3.14, *p* = < 0.01) and HDL levels (t = 4.78, *p* = < 0.01) were found to be higher in women than men. No further statistical significant differences between the two genders were identified in univariate analyses.

The first intermediate binary logistic regression model with clinical qualitative variables as predictors (Table 3) was found to be reliable (Omnibus test: χ^2^ = 23.89, *p* < 0.01; Hosmer and Lemeshow test: χ^2^ = 10.77, *p* = 0.22) allowing for a correct classification of 75% of cases. The model confirmed that female bipolar patients versus male patients were more likely to have psychiatric comorbidity (*p* < 0.01).

The second intermediate binary logistic regression model with clinical continuous variables as predictors (Table 4) proved to be reliable (Hosmer and Lemeshow test: χ^2^ = 11.09, *p* = 0.14), allowing for a correct classification of 89.3% of cases. The model confirmed that female bipolar patients had better functioning at the beginning of hospitalization (*p* = 0.05).

The third intermediate binary logistic regression model with biochemical parameters as predictors (Table 5) proved to be reliable (Omnibus test: χ^2^ = 93.82, *p* < 0.01; Hosmer and Lemeshow test: χ^2^ = 8.71, *p* = 0.37), allowing for a correct classification of 86.0% of cases. The model confirmed that female bipolar patients had fewer red blood cells (*p* = 0.05), lower creatinine plasma levels (*p* < 0.01) and higher total cholesterol (*p* < 0.01).

The final binary logistic regression model was reliable (Omnibus test: χ^2^ = 14.41, *p* = 0.04; Hosmer and Lemeshow test: χ^2^ = 4.87, *p* = 0.77) allowing for a correct classification of 87.7% of cases (Table 6). Red blood cells proved to be lower in female versus male bipolar patients (*p* = 0.01).

## 4. Discussion

Our data show several gender differences in the clinical variables and biochemical parameters of bipolar patients, with some being confirmed by logistic regression models.

Regarding clinical variables, some findings mirror the previous literature, such as the higher frequency in males versus females in regard to substance use disorders [19,34] or the later age of onset in bipolar females compared to males [35,36]. Similarly, previous research reported more suicide attempts in bipolar women than men [37], as well as more frequent tobacco smoking in males versus females [38].

Although our sample did not show a higher frequency of rapid cycling, as previously reported in the literature [8,39], we found significant differences in the number of mood episodes in the year preceding hospitalization, which appears to be higher in females regardless of the episode polarity. These findings partially contrast with other studies reporting a higher number of manic episodes in males [40,41,42]. However, it must be considered that, if lifetime episodes are counted and not just recent ones, women experience more depressive episodes than men. Greater clinical destabilization, especially in the hypomanic or manic sense, could more frequently result in hospital admission for women due to their greater propensity to seek psychiatric help in cases of mental health problems [43].

In our sample, a higher frequency of lifetime psychotic symptoms was identified in males versus females, although a large meta-analysis found no substantial differences between the two genders [44]. This could be due to the specific characteristics of our sample, such as the greater frequency of substance use disorders in males. In addition, male hormones could contribute to a greater susceptibility to the development of psychotic symptoms, but this aspect has not yet been fully clarified by the existing literature [45].

In agreement with our findings, some authors in different cultural contexts [42,46] noted that bipolar women are more frequently affected by psychiatric and medical comorbidities than men. Bipolar women would seem more susceptible to endocrine and metabolic problems than men [47,48], especially after the menopause. Hormonal changes related to the menstrual cycle and menopause probably play an important role in explaining differences between the sexes in terms of medical and psychiatric comorbidity [49]. This would also explain the later age of onset in women and the fact that BD in females is often preceded by anxiety or eating disorders [50].

Globally, our data indicate that women hospitalized for BD (than men) are more severe in terms of rating scale scores [51], achieve treatment remission less frequently and experience medical and psychiatric comorbidities more often. Despite these unfavourable clinical factors, women have better social functioning, as indicated by higher GAF scores. This apparent contradiction could be explained by the pro-cognitive effect of oestrogens [52]; of note, several authors reported more cognitive impairment in male versus female bipolar patients, especially in the first stages of illness [53].

Regarding biochemical parameters, several differences between genders were detected in our data, some of which reflect the diversity in the general healthy population (e.g., hemoglobin or red blood cell count) [54].

In bipolar patients, changes in WBC count were correlated with greater symptom severity, particularly in males [55], with higher values being found, especially in manic phases [56]. In another cohort of bipolar patients, no gender differences in leukocyte count were found [20], in opposition to our sample, where females showed lower total leukocytes compared to males. It has to be noted that no significant difference exists between the male and female values of leukocytes or neutrophils in the general population [54,57]. Our result may be explained by different factors, including the fact that male bipolar patients experience more substance use disorders than females. A recent study reported that abnormally increased white blood cell counts can be observed even in young adults with family history for substance use disorders [58].

An interesting result is represented by the MCV, which is significantly higher in women than in men in our results, when, in the general population, there are either no differences [54] or slightly higher values in men [59]. It is the first time that a similar result was found in bipolar patients: a study conducted in China observed an association between MCV and severity of illness, but only for males [60].

Regarding blood creatinine [61], AST, ALT, bilirubin, serum albumin [62], uric acid [63] and CPK values [64] were all significantly higher in males compared to females in our sample, mirroring the general population. In contrast, as already demonstrated in samples of patients suffering from schizophrenia [33] and major depressive disorder [65], women hospitalized for bipolar disorder have higher cholesterol levels than men, confirming the vulnerability of the female gender for metabolic disorders.

A series of clinical considerations can be made based on our results. Bipolar men tend to be smokers and substance abusers. It would therefore be appropriate to adjust the doses of the drugs, choosing the compounds with a lower cardiac impact or hepatic metabolism [66]. Clinicians could plan specific psychoeducational interventions for bipolar men to avoid substance addictions or cognitive rehabilitation to improve treatment adherence and social functioning [67]. In the case of women, given the greater clinical instability and the frequent psychiatric comorbidity, a combined pharmacological treatment may be necessary while balancing the risk of further metabolic problems [68]. In this sense, compounds with low metabolic impact should be preferred [68]. Lithium is an agent used to treat depression and can help in treating suicidal tendencies [69]. It would be important for bipolar women to plan medical visits to identify metabolic pathologies or monitor any current medical conditions.

## 5. Conclusions

Compared to men, at the time of hospitalization for BD, women show a greater clinical severity in terms of more prominent symptoms and more frequent medical/psychiatric comorbidity as well as more metabolic vulnerability. Despite this, female bipolar patients have better global functioning than men. Our data confirm that treatment should be tailored according to gender with targeted pharmacotherapy and psychosocial interventions. Bipolar women would benefit from a treatment that has less impact on metabolic aspects, while, in regard to men, attention must be paid to substance misuse.

The strength of this study is represented by the fact that a large number of clinical variables and biochemical parameters were analyzed simultaneously in a naturalistic context. These results can provide insights into clinical practice, with, for example, greater attention being paid to the impact of substance abuse in men and medical/psychiatric comorbidity in women in the context of precision psychiatry. In addition, it is the first time that gender differences in MCV has been reported in a sample of bipolar patients: this finding has to be confirmed by other studies, analyzing the factors that can have contributed to this (such as a poorer diet in women, which resulted more frequently in overweight than men in our sample).

This study has the following limitations: (1) although no gender differences were identified in the last main pharmacological treatment, the possibility that the last pharmacotherapy may have influenced clinical and biological parameters cannot be excluded; (2) biochemical parameters may have been affected by substance use disorders or medical comorbidities; (3) the retrospective design of the study may make information less accurate; (4) the lack of a follow-up; (5) some data are missing because some parameters were not routinely collected at the admission of patients in one or both centres; (6) some patients with BD were identified according to the DSM-IV-TR criteria, and some subjects with antidepressant-induced hypomania or mania may not have been recruited (although it must be considered that the majority of patients were recruited after the release of the DSM-5 and that the last hospitalization was taken into account); (7) GAF scores could have been influenced by other factors such as psychiatric comorbidity.

Further studies are needed to confirm gender differences in BD and translate results in clinical practice, contributing to the implementation of precision medicine.

## Figures and Tables

**Table 1 brainsci-15-00214-t001:** Qualitative variables of the total sample and of the two groups divided according to gender.

Variables		Total Sample N = 672	Males N = 277 (41.2%)	Females N = 395 (58.8%)	χ^2^	OR (95% CI)	*p*
Type of current episode	Manic	514 (76.5%)	216 (78.0%)	298 (75.4%)	0.58	1.15 (0.80–1.66)	0.46
	Depressive	158 (23.5%)	61 (22.0%)	97 (24.6%)			
Current presence of mixed features Missing *n* = 4	Yes	191 (28.6%)	65 (23.7%)	126 (32.0%)	5.40	1.51 (1.07–2.15)	0.24
	No	477 (71.4%)	209 (76.3%)	268 (68.0%)			
Lifetime presence of mixed features Missing *n* = 232	Yes	247 (56.1%)	92 (52.3%)	155 (58.7%)	1.78	1.30 (0.88–1.91)	0.20
	No	193 (43.9%)	84 (47.7%)	109 (41.3%)			
Lifetime presence of psychotic symptoms Missing *n* = 5	Yes	429 (64.3%)	189 (69.0%)	240 (61.1%)	4.40	0.71 (0.51–0.98)	**0.04**
	No	238 (35.7%)	85 (31.0%)	153 (38.9%)			
Type of Bipolar Disorder Missing *n* = 48	1	614 (98.4%)	250 (99.2%)	364 (97.8%)	1.75	2.75 (0.58–13.05)	0.22
	2	10 (1.6%)	2 (0.8%)	8 (2.2%)			
Lifetime presence of rapid cycling Missing *n* = 204	Yes	64 (13.7%)	26 (16.9%)	38 (12.1%)	2.00	0.68 (0.40–1.16)	0.20
	No	404 (86.3%)	128 (83.1%)	276 (87.9%)			
Lifetime presence of seasonality Missing *n* = 266	Yes	28 (6.9%)	7 (5.9%)	21 (7.3%)	0.27	1.26 (0.52–3.06)	0.67
	No	378 (93.1%)	112 (94.1%)	266 (92.7%)			
Family history of psychiatric disorders Missing *n* = 260	Yes	204 (49.5%)	74 (48.1%)	130 (50.4%)	0.21	1.10 (0.74–1.64)	0.68
	No	208 (50.5%)	80 (51.9%)	128 (49.6%)			
Multiple family history of psychiatric disorders Missing *n* = 271	Yes	123 (30.7%)	47 (31.1%)	76 (30.4%)	0.02	0.97 (0.62–1.50)	0.91
	No	278 (69.3%)	104 (68.9%)	174 (69.6%)			
Presence of lifetime substance use disorders Missing *n* = 60	Yes	198 (32.4%)	103 (41.0%)	95 (26.3%)	14.66	0.51 (0.36–0.72)	**<0.01**
	No	414 (67.6%)	148 (59.0%)	266 (73.7%)			
Presence of lifetime alcohol use disorders Missing *n* = 79	Yes	117 (19.7%)	52 (22.2%)	65 (18.1%)	1.52	0.77 (0.51–1.17)	0.25
	No	476 (80.3%)	182 (77.8%)	294 (81.9%)			
Presence of lifetime multiple substance use disorders Missing *n* = 71	Yes	79 (13.1%)	40 (16.4%)	39 (10.9%)	3.80	0.63 (0.39–1.01)	0.07
	No	522 (86.9%)	204 (83.6%)	318 (89.1%)			
Current tobacco smoking Missing *n* = 227	Yes	244 (54.8%)	121 (65.8%)	123 (47.1%)	15.13	0.46 (0.31–0.69)	**<0.01**
	No	201 (45.2%)	63 (34.2%)	138 (52.9%)			
Type of the last mood episode Missing *n* = 307	Current first episode	3 (0.8%)	2 (1.5%)	1 (0.5%)	4.53	NA	0.21
	Depressive	153 (41.8%)	64 (46.7%)	89 (39.0%)			
	Manic	152 (41.4%)	48 (35.0%)	103 (45.2%)			
	Hypomanic	58 (15.8)	23 (16.8%)	35 (15.3%)			
Administration of poly-therapy during the last mood episode Missing *n* = 269	Yes	173 (42.9%)	70 (48.3%)	103 (39.9%)	2.64	0.71 (0.47–1.07)	0.12
	No	230 (57.1%)	75 (51.7%)	155 (60.1%)			
Comorbid personality disorder Missing *n* = 187	Yes	63 (13.0%)	14 (7.8%)	49 (16.1%)	6.88	2.27 (1.21–4.24)	**0.01**
	No	422 (87.0%)	166 (92.2%)	256 (83.9%)			
Psychiatric comorbidity Missing *n* = 487	Yes	77 (41.6%)	16 (24.2%)	61 (51.3%)	12.75	3.29 (1.69–6.41)	**<0.01**
	No	108 (58.4%)	50 (75.8%)	58 (48.8%)			
Previous suicide attempts Missing *n* = 154	Yes	118 (22.8%)	37 (18.7%)	81 (25.3%)	3.05	1.48 (0.95–2.28)	0.09
	No	400 (77.2%)	161 (81.3%)	239 (74.7%)			
Medical comorbidity Missing *n* = 233	Yes	188 (42.8%)	38 (22.6%)	150 (55.4%)	45.38	4.24 (2.75–6.54)	**<0.01**
	No	251 (57.2%)	130 (77.4%)	121 (44.6%)			
Medical policomorbidity Missing *n* = 249	Yes	119 (28.1%)	16 (10.3%)	103 (38.6%)	39.06	5.50 (3.10–9.75)	**<0.01**
	No	304 (71.9%)	140 (89.7%)	164 (61.4%)			
Obstetrical complications Missing *n* = 190	Yes	9 (1.9%)	4 (2.2%)	5 (1.7%)	0.15	0.77 (0.20–2.99)	0.74
	No	473 (98.1%)	180 (97.8%)	293 (98.3%)			
Comorbidity with thyroid diseases Missing *n* = 174	No	420 (84.3%)	184 (96.8%)	236 (76.6%)	36.60	NA	**<0.01**
	Hypo-thyroidism	70 (14.1%)	6 (3.2%)	64 (20.8%)			
	Hyper-thyroidism	8 (1.6%)	0 (0.0%)	8 (2.6%)			
Comorbidity with diabetes Missing *n* = 175	Yes	47 (9.5%)	17 (9.0%)	30 (9.7%)	0.61	1.08 (0.58–2.02)	0.88
	No	450 (90.5%)	171 (91.0%)	279 (90.3%)			
Comorbidity with hypercholesterolemia Missing *n* = 238	Yes	106 (24.4%)	19 (11.6%)	87 (32.2%)	23.54	3.63 (2.11–5.24)	**<0.01**
	No	328 (75.6%)	145 (88.4%)	183 (67.8%)			
Comorbidity with obesity Missing *n* = 263	Yes	31 (7.6%)	5 (3.2%)	26 (11.4%)	6.75	3.42 (1.29–9.11)	**0.01**
	No	378 (92.4%)	150 (96.8%)	228 (88.6%)			
Achievement of treatment response in the current episode Missing *n* = 163	Yes	474 (93.1%)	181 (93.3%)	293 (93.0%)	0.02	0.96 (0.47–1.95)	1.00
	No	35 (6.9%)	13 (6.7%)	22 (7.0%)			
Achievement of remission in the current episode Missing *n* = 163	Yes	346 (68.0%)	146 (74.9%)	200 (63.7%)	6.90	0.59 (0.40–0.88)	**0.01**
	No	163 (32.0%)	49 (25.1%)	114 (36.3%)			
Current treatment with statins Missing *n* = 222	Yes	23 (5.1%)	12 (8.1%)	11 (3.6%)	4.08	0.43 (0.18–1.00)	0.07
	No	427 (94.9%)	136 (91.9%)	291 (96.4%)			
Current treatment with levothyroxine Missing *n* = 297	Yes	32 (8.5%)	1 (0.7%)	31 (13.5%)	18.64	22.43 (3.03–166.22)	**<0.01**
	No	343 (91.5%)	144 (99.3%)	199 (86.5%)			

Legend: CI: confidence interval; χ^2^: chi-square; NA: not applicable; OR: odds ratio; *p*: *p* value. Frequencies with percentages in brackets are reported. Results in bold are statistically significant *p* (≤0.05). OR refers to females versus males.

**Table 2 brainsci-15-00214-t002:** Continuous variables of the total sample and of the two groups divided according to gender.

Variables	Total Sample N = 672	Males N = 277 (41.2%)	Females N = 395 (58.8%)	t	*p*
Duration of hospitalization (days) Missing *n* = 152	12.73 ± 8.30	12.99 ± 8.96	12.56 ± 7.87	0.57	0.56
Age at admission (years)	56.96 ± 14.29	44.58 ± 14.31	47.84 ± 14.13	2.93	**<0.01**
Age at illness onset (years) Missing *n* = 99	28.97 ± 11.23	27.70 ± 10.76	29.75 ± 11.45	2.13	**0.03**
Duration of Untreated Illness (DUI) (years) Missing *n* = 246	3.02 ± 5.38	2.51 ± 4.71	3.25 ± 5.66	1.30	0.19
Duration of illness (DI) (years) Missing *n* = 99	17.74 ± 12.94	16.89 ± 12.52	18.27 ± 13.20	1.24	0.21
Number of previous psychiatric hospitalizations Missing *n* = 149	3.36 ± 4.55	3.21 ± 4.32	3.46 ± 4.71	0.62	0.54
Number of previous acute episodes Missing *n* = 199	6.10 ± 5.41	5.64 ± 4.42	6.38 ± 5.94	1.46	0.15
Number of previous manic episodes Missing *n* = 198	2.51 ± 3.41	2.34 ± 2.19	2.61 ± 3.98	0.83	0.41
Number of previous hypomanic episodes Missing *n* = 199	1.34 ± 1.90	1.26 ± 1.89	1.38 ± 1.90	0.69	0.49
Number of previous depressive episodes Missing *n* = 200	2.23 ± 2.04	2.00 ± 1.96	2.38 ± 2.07	1.99	**0.04**
Number of previous suicide attempts Missing *n* = 163	0.37 ± 0.86	0.28 ± 0.69	0.43 ± 0.95	1.94	**0.05**
Number of manic episodes in the last year Missing *n* = 240	0.98 ± 0.67	0.81 ± 0.73	1.07 ± 0.62	3.82	**<0.01**
Number of hypomanic episodes in the last year Missing *n* = 239	0.21 ± 0.58	0.15 ± 0.45	0.25 ± 0.63	1.98	**0.05**
Number of depressive episodes in the last year Missing *n* = 241	0.33 ± 0.66	0.26 ± 0.54	0.37 ± 0.71	1.81	0.07
Cumulative number of mood episodes in the last year Missing *n* = 239	1.49 ± 1.03	1.14 ± 0.88	1.68 ± 1.05	5.69	**<0.01**
Number of previous substance-induced episodes Missing *n* = 286	0.13 ± 0.46	0.14 ± 0.38	0.12 ± 0.49	0.38	0.71
Current YMRS score Missing *n* = 163	20.45 ± 10.56	19.07 ± 9.88	21.32 ± 10.90	2.36	**0.02**
Current HAM-D score Missing *n* = 506	14.56 ± 6.61	13.07 ± 5.90	15.43 ± 6.87	2.25	**0.03**
Current MADRS score Missing *n* = 551	22.53 ± 8.65	23.91 ± 8.73	21.77 ± 8.56	1.31	0.19
Current BPRS score Missing *n* = 142	40.94 ± 9.19	39.63 ± 9.50	41.81 ± 8.88	2.70	**<0.01**
Current HAM-A score Missing *n* = 569	8.95 ± 4.36	8.43 ± 4.09	9.24 ± 4.50	0.91	0.37
Current GAF score (Global Assessment of Functioning) Missing *n* = 287	56.96 ± 14.33	53.59 ± 13.67	57.82 ± 14.38	2.34	**0.02**
BMI (kg/m^2^) Missing *n* = 523	25.13 ± 5.42	24.87 ± 4.17	25.36 ± 6.36	0.56	0.58
Cigarettes/day Missing *n* = 348	9.35 ± 12.07	13.16 ± 14.05	7.25 ± 10.28	3.96	**<0.01**
Sodium (mmol/L) Missing *n* = 312	141.73 ± 2.69	141.77 ± 2.46	141.70 ± 2.84	0.21	0.84
Potassium (mmol/L) Missing *n* = 316	4.27 ± 2.41	4.49 ± 3.77	4.13 ± 0.41	1.40	0.16
White Blood Cells (10⁹/L) Missing *n* = 180	7.70 ± 2.64	8.10 ± 2.70	7.43 ± 2.56	2.81	**0.01**
Red Blood Cells (10⁶/mm^3^) Missing *n* = 170	4.60 ± 0.55	4.91 ± 0.53	4.38 ± 0.47	11.81	**<0.01**
Hemoglobin (g/dL) Missing *n* = 169	13.65 ± 1.68	14.74 ± 1.47	12.89 ± 1.38	14.34	**<0.01**
MCV (fL) Missing *n* = 313	86.47 ± 9.04	85.06 ± 12.38	87.33 ± 6.10	2.32	**0.02**
Platelets (10^9^/L) Missing *n* = 317	246.17 ± 67.62	239.01 ± 64.80	250.56 ± 69.08	1.57	0.12
MPV (fL) Missing *n* = 318	11.02 ± 4.50	10.76 ± 0.85	11.19 ± 5.66	0.87	0.39
Neutrophils (10⁹/L) Missing *n* = 269	4.08 ± 2.49	4.49 ± 2.49	3.80 ± 2.45	2.73	**0.01**
Lymphocytes (10⁹/L) Missing *n* = 268	2.17 ± 0.75	2.17 ± 0.77	2.18 ± 0.73	0.09	0.93
Blood Glucose (mg/dL) Missing *n* = 192	94.41 ± 26.73	95.76 ± 26.87	93.41 ± 26.63	0.95	0.34
Urea (mg/dL) Missing *n* = 366	29.81 ± 16.57	30.63 ± 12.51	29.20 ± 19.04	0.75	0.46
Creatinine (mg/dL) Missing *n* = 179	0.85 ± 0.34	0.94 ± 0.21	0.79 ± 0.40	4.88	**<0.01**
Uric acid (mg/dL) Missing *n* = 328	5.19 ± 1.90	5.64 ± 1.39	4.91 ± 2.12	3.49	**<0.01**
AST (mU/mL) Missing *n* = 348	26.74 ± 37.09	32.71 ± 54.38	22.26 ± 12.24	2.22	**0.03**
ALT (mU/mL) Missing *n* = 272	24.68 ± 18.05	29.57 ± 20.28	20.69 ± 14.89	4.89	**<0.01**
GGT (U/L) Missing *n* = 290	24.92 ± 29.30	29.34 ± 24.35	21.35 ± 32.40	0.17	0.08
Bilirubin (mg/dL) Missing *n* = 202	0.56 ± 0.40	0.66 ± 0.50	0.49 ± 0.30	4.24	**<0.01**
Plasmatic protein (g/dL) Missing *n* = 343	6.60 ± 0.57	6.67 ± 0.58	6.54 ± 0.56	2.07	**0.04**
Albumin (g/dL) Missing *n* = 332	4.26 ± 0.43	4.36 ± 0.42	4.19 ± 0.42	3.51	**<0.01**
LDH (mU/mL) Missing *n* = 426	208.65 ± 95.69	204.36 ± 111.23	211.64 ± 83.44	0.59	0.56
CPK (U/L) Missing *n* = 336	215.65 ± 333.85	302.81 ± 429.24	163.36 ± 247.24	3.33	**<0.01**
PCHE (U/L) Missing *n* = 438	7281.97 ± 1727.82	7541.96 ± 1744.83	7104.27 ± 1699.54	1.91	0.06
Total cholesterol (mg/dL) Missing *n* = 222	176.25 ± 41.06	168.98 ± 41.38	181.22 ± 40.17	3.14	**<0.01**
HDL (mg/dL) Missing *n* = 492	52.47 ± 16.03	46.77 ± 15.45	57.56 ± 14.85	4.78	**<0.01**
LDL (mg/dL) Missing *n* = 508	105.61 ± 35.67	105.22 ± 36.21	106.00 ± 35.35	0.14	0.89
Triglycerides (mg/dL) Missing *n* = 478	115.84 ± 73.32	118.74 ± 62.81	113.38 ± 81.40	0.51	0.61
Blood Iron (µg/dL) Missing *n* = 462	83.21 ± 39.20	84.03 ± 37.08	82.72 ± 40.60	0.23	0.82
TSH (mU/L) Missing *n* = 302	2.14 ± 2.25	1.88 ± 1.59	2.28 ± 2.54	1.87	0.06
PCR (mg/L) Missing *n* = 531	1.31 ± 2.85	1.23 ± 2.10	1.37 ± 3.25	0.28	0.78
NLR Missing *n* = 269	2.16 ± 1.77	2.31 ± 1.66	2.04 ± 1.83	1.55	0.12
PLR Missing *n* = 328	125.80 ± 53.83	124.38 ± 54.20	126.69 ± 53.71	0.39	0.70
AST/ALT Missing *n* = 350	1.20 ± 0.78	1.17 ± 1.08	1.23 ± 0.42	0.65	0.52
Sodium/Potassium Ratio Missing *n* = 317	34.41 ± 3.82	34.01 ± 4.24	34.68 ± 3.50	1.63	0.11

Legend: ALT: alanine transaminase; AST: aspartate transaminase; BMI: Body Mass Index; BPRS: Brief Psychiatric Rating Scale; CPK: creatine phosphokinase; GAF: Global Assessment of Functioning; GGT: gamma-glutamyl transferase; HAM-A: Hamilton; Anxiety Rating Scale; HAM-D: Hamilton Depression Rating Scale; HDL: high-density lipoproteins; LDH: lactate dehydrogenase; LDL: low-density lipoproteins; MADRS: Montgomery and Asberg Depression Rating Scale; MCV: mean corpuscular volume; MPV: mean platelet volume; NLR: Neutrophils/Lymphocyte Ratio; *p*: *p* value; PCHE: pseudocholinesterase; PCR: C-reactive protein; PLR: Platelet/Lymphocyte Ratio; t: Student’s *t*; TSH: thyroid-stimulating hormone; YMRS: Young Mania Rating Scale. Results in bold are statistically significant *p* (≤0.05). Means ± standard deviations are reported.

**Table 3 brainsci-15-00214-t003:** Summary of the statistics of the binary regression model for clinical qualitative variables.

Variables	B	S.E.	Wald	*p*	EXP(B)	95% CI for EXP(B)
Psychiatric comorbidity	1.67	0.53	9.85	**<0.01**	5.31	**1.87–15.06**
Achievement of remission in the current episode	−0.83	0.51	2.67	0.10	0.44	0.16–1.18
Current treatment with statins	0.97	0.61	2.52	0.11	2.65	0.80–8.80
Current presence of mixed features	0.58	0.47	1.49	0.22	1.78	0.71–4.50
Lifetime presence of psychotic symptoms	−0.77	0.48	2.54	0.11	0.47	0.18–1.19
Presence of lifetime substance use disorders	−0.32	0.48	0.45	0.50	0.73	0.28–1.86
Current tobacco smoking	−0.28	0.49	0.33	0.57	0.76	0.29–1.96
Comorbid personality disorder	0.58	0.68	0.72	0.40	1.79	0.47–6.83
Comorbidity with obesity	0.26	0.89	0.08	0.77	1.30	0.23–7.37

Legend: In this analysis, the dependent variable was female gender versus male one. Results in bold are statistically significant *p*. B = regression coefficient; CI = confidence interval; EXP(B) = B exponential; *p* = *p* value; S.E. = standard error of B; Wald = Wald statistics.

**Table 4 brainsci-15-00214-t004:** Summary of the statistics of binary regression model for clinical continuous variables.

Variables	B	S.E.	Wald	*p*	EXP(B)	95% CI for EXP(B)
Age at illness onset (years)	0.06	0.08	0.66	0.42	1.06	0.92–1.24
Current GAF score	0.15	0.08	3.98	**0.05**	1.16	1.00–1.35
Cigarettes/day	−0.05	0.04	1.51	0.22	0.95	0.88–1.03
Number of previous depressive episodes	−0.39	0.25	2.35	0.13	0.68	0.41–1.11
Number of previous suicide attempts	1.16	0.77	2.28	0.13	3.18	0.71–14.36
Cumulative number of mood episodes in the last year	0.30	0.73	0.17	0.68	1.35	0.33–5.59
Current YMRS score	−0.12	0.14	0.77	0.38	0.89	0.68–1.16
Current MADRS score	0.02	0.10	0.03	0.86	1.02	0.83–1.25
Current BPRS score	0.11	0.18	0.39	0.53	1.12	0.79–1.58

Legend: In this analysis, the dependent variable was a female gender versus male one. Results in bold are statistically significant *p*. B = regression coefficient; BPRS = Brief Psychiatric Rating Scale; CI = confidence interval; EXP(B) = B exponential; GAF = Global Assessment of Functioning; MADRS = Montgomery and Asberg Depression Rating Scale; YMRS = Young Mania Rating Scale. *p* = *p* value; S.E. = standard error of B; Wald = Wald statistics.

**Table 5 brainsci-15-00214-t005:** Summary of the statistics of the binary regression model for biochemical parameters.

Variables	B	S.E.	Wald	*p*	EXP(B)	95% CI for EXP(B)
Red Blood Cells (10^6^/mm^3^)	−3.53	1.80	3.81	**0.05**	0.03	0.01–1.01
Hemoglobin (g/dL)	0.02	0.56	0.01	0.97	1.02	0.34–3.05
MCV (fl)	−0.16	0.11	2.12	0.15	0.86	0.70–1.06
Neutrophils (10^9^/L)	−0.29	0.16	3.43	0.06	0.75	0.55–1.02
Urea (mg/dL)	−0.03	0.19	0.02	0.89	0.98	0.68–1.40
Creatinine (mg/dL)	−6.39	1.56	16.72	**<0.01**	0.01	0.01–0.04
AST (mU/mL)	−0.01	0.04	0.06	0.80	1.00	0.92–1.07
ALT (mU/mL)	−0.03	0.03	1.05	0.30	0.97	0.92–1.03
GGT (U/L)	−0.01	0.01	0.74	0.39	0.99	0.98–1.01
Bilirubin (mg/dL)	0.66	0.82	0.65	0.42	1.93	0.39–9.63
Plasmatic protein (g/dL)	0.66	0.73	0.81	0.37	1.93	0.46–8.06
Albumin (g/dL)	0.88	1.13	0.61	0.43	2.41	0.27–21.85
CPK (U/L)	−0.01	0.01	1.67	0.20	1.00	0.99–1.00
Total cholesterol (mg/dL)	0.02	0.01	8.34	**<0.01**	1.02	1.02–1.01

Legend: In this analysis the dependent variable was female gender versus male one. Results in bold are statistically significant *p*. ALT = alanine transaminases; AST = aspartate transaminase; B = regression coefficient; CI = confidence interval; CPK = creatine phosphokinase; EXP(B) = B exponential; GGT = gamma-glutamyl transferase; MCV = mean corpuscular volume; *p* = *p* value; S.E. = standard error of B; Wald = Wald statistics.

**Table 6 brainsci-15-00214-t006:** Final binary logistic regression model.

Variables	B	S.E.	Wald	*p*	EXP(B)	95% CI for OR
Total cholesterol	0.01	0.01	0.49	0.48	1.01	0.99–1.03
GAF score	0.01	0.02	0.36	0.55	1.01	0.97–1.06
Psychiatric comorbidity	0.32	0.74	0.36	0.55	1.38	0.32–5.86
Comorbidity with thyroid diseases	1.74	1.30	1.79	0.18	5.71	0.44–73.39
Red Blood Cells (10^6^/mm^3^)	−2.10	0.84	6.30	**0.01**	0.12	**0.02–0.63**
Neutrophils (10⁹/L)	0.12	0.18	0.42	0.52	1.12	0.79–1.60
Creatinine (mg/dL)	−0.92	1.43	0.41	0.52	0.40	0.02–6.63

In this analysis, the dependent variable was the female gender versus the male gender. Results in bold are statistically significant *p*. B = regression coefficient; CI = confidence interval; EXP(B) = B exponential; GAF = Global Assessment of Functioning; *p* = *p* value; S.E. = standard error of B; Wald = Wald statistics.

## Data Availability

The original contributions presented in this study are included in the article. Further inquiries can be directed to the corresponding author.

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
