# Peer review of "Gender Differences in Clinical and Biochemical Variables of Patients Affected by Bipolar Disorder"

_brainsci, 2025, doi:10.3390/brainsci15020214_

Round 1

Reviewer 1 Report

Comments and Suggestions for Authors

The article addresses sex differences in clinical and biochemical parameters in patients with bipolar disorder. The article is well prepared, with a sufficient introduction and adequate references, with a clear methodology in terms of the variables taken and a remarkable statistical methodology. However, there are a few points to improve so that it is ready for possible publication.

As for the Abstract, it is brief with all the above done since it is not mentioned the results of the univariate analysis, as well as in the wording of the results there is little significant data in the metabolic and clinical aspects that do not support something as relative as "seem to keep a better functionin".

As for the wording of the results, it is difficult to analyze them because there are more words in the tables than were observed in the description of the data obtained. What are the relevant metabolic and clinical parameters in the univariate analysis, and which are relevant in the multivariate analysis?

The discussion is short and concise because there are so many results and statistical weight that many comment on it as previously reported. These expressions give less support to originality and the work done by the authors. It is suggested that this section be rewritten, emphasizing the strengths of the study. As for the wording of the results, it is difficult to analyze them because there are more words in the tables than were observed in the description of the data obtained. What are the relevant metabolic and clinical parameters in the univariate analysis, and which are relevant in the multivariate analysis? The discussion is short and concise because, having so many results and statistical weight, many comment on it as previously reported. These expressions give less support to originality and the work done by the authors. It is suggested that this section be rewritten, emphasizing the strengths of the study

Author Response

First of all we would like to thank the reviewer for the precious suggestions which helped us to improve the present manuscript.

1)The article addresses sex differences in clinical and biochemical parameters in patients with bipolar disorder. The article is well prepared, with a sufficient introduction and adequate references, with a clear methodology in terms of the variables taken and a remarkable statistical methodology. However, there are a few points to improve so that it is ready for possible publication.

We thank the reviewer for the appreciation. 

 2) As for the Abstract, it is brief with all the above done since it is not mentioned the results of the univariate analysis, as well as in the wording of the results there is little significant data in the metabolic and clinical aspects that do not support something as relative as "seem to keep a better functioning"

Many results were not reported because of the 200-word limit specified by the journal. The finding of better overall functioning of female versus male patients would be supported by the significant difference on the GAF scale scores.

3) As for the wording of the results, it is difficult to analyze them because there are more words in the tables than were observed in the description of the data obtained. What are the relevant metabolic and clinical parameters in the univariate analysis, and which are relevant in the multivariate analysis?

Thanks for your observation. We-reorganized the paragraph about the presentation of results, grouping the findings in three sections (clinical variables, rating scale scores and biochemical parameters). Furthermore only statistically significant variables in univariate analyses as well as in binary logistic regression models were reported.

4) The discussion is short and concise because, having so many results and statistical weight, many comment on it as previously reported. These expressions give less support to originality and the work done by the authors. It is suggested that this section be rewritten, emphasizing the strengths of the study

Thanks for your useful suggestion. We explained the strengths of the study and added a part on clinical applications, in order to make the results of our study more usable and original.In addition, it is the first time that gender differences in MCV is reported in a sample of bipolar patients: this finding has to be confirmed by other studies, analyzing the factors that can have contribute to this (such as a poorer diet in women that resulted more frequently in overweight than men in our sample). All these aspects were remarked upon in the discussion.

Reviewer 2 Report

Comments and Suggestions for Authors

This manuscript addresses an interesting topic, i.e., the gender differences in relation to clinical and biochemical variables of patients diagnosed with bipolar disorders (BDs). The results of this study may have important clinical and therapeutic implications. Please refer to the following observations:

Line 79- which edition(s) of the DSM was/were included? In the latest editions of the DSM, BD induced by substances or general medical conditions may be diagnosed, so were these entities also considered by the present study? Exclusion criteria (2) does not cover all these cases. It is not clear if all psychiatric comorbidities were allowed (lines 82-83), and if so, how their impact on the GAF scores could be mitigated. Line 85- Was any list of pharmacological agents that led to the exclusion of the cases, based on suspected causal effect on the BD?

Line 94- the date of the Local Ethical Committee approval is required after its number;

Line 132- how was the onset of the illness determined, e.g., by self-reporting, by a caregiver’s testimony, etc.;

Table 1- how was the personality disorder assessed? This aspect is not mentioned in the line 116, either. What does “polytherapy” mean in this context, i.e., combinations of moodstabilizers or any drug association, like a moodstabilizer and an add-on agent, for example?

The impact of pharmacological agents on metabolic parameters is difficult to assess based on the presented data. Mirtazapine, olanzapine, and tricyclic antidepressants are associated with more frequent weight gain, dyslipidemia, and diabetes than SSRIs, SNRIs, or aripiprazole, but it is not clear how these agents were distributed across genders.

Author Response

First of all we would like to thank the reviewer for the precious suggestions which helped us to improve the present manuscript.

1) This manuscript addresses an interesting topic, i.e., the gender differences in relation to clinical and biochemical variables of patients diagnosed with bipolar disorders (BDs). The results of this study may have important clinical and therapeutic implications. 

Thanks for the appreciation.

Please refer to the following observations

2) Line 79- which edition(s) of the DSM was/were included? In the latest editions of the DSM, BD induced by substances or general medical conditions may be diagnosed, so were these entities also considered by the present study? Exclusion criteria (2) does not cover all these cases. It is not clear if all psychiatric comorbidities were allowed (lines 82-83), and if so, how their impact on the GAF scores could be mitigated. Line 85- Was any list of pharmacological agents that led to the exclusion of the cases, based on suspected causal effect on the BD?

Thanks for your observation. The patients were hospitalized from 2008 to 2023 so that DSM-IV-TR (text revision) or DSM-5 criteria were applied. Some patients with BD were identified according to the DSM-IV-TR criteria, then few subjects with antidepressant-induced hypomania or mania may not have been recruited (although it must be considered that the majority of patients were recruited after the release of the DSM-5 and the last hospitalization was considered). All this information was added in the methods and study limitations. In the methods, we remarked the fact that the information of the presence of psychiatric comorbidity was collected. If the subjects satisfied the criteria for other psychiatric disorders, BD represented the main diagnosis (associated with more social dysfunction). GAF scores can have been influenced by other factors such as psychiatric comorbidity: this is a strength of the results of the study as psychiatric comorbidity resulted to be more frequent in female patients despite higher GAF scores. With regard to pharmacological agents at the time of hospitalization only few patients received an antidepressant as main pharmacological treatment:  9 with Selective Serotonin Reuptake Inhibitors (SSRIs), 11 with Selective Serotonin Noradrenaline Reuptake Inhibitors (SNRIs), 4 with vortioxetine, 2 with mirtazapine, 2 with trazodone, 2 with tricyclic antidepressants. As mentioned above, most of the patients were hospitalized after the DSM-5 release so that very few patients might be excluded as a result of antidepressant-induced hypomania or mania. We excluded the patients that during the hospitalization were in current treatment with steroids, levetiracetam, interferon, efavirenz: we detailed this information in the methods.

3) Line 94- the date of the Local Ethical Committee approval is required after its number;

We added it as you requested.

4) Line 132- how was the onset of the illness determined, e.g., by self-reporting, by a caregiver’s testimony, etc.;

We specified how this information was collected. 

5) Table 1- how was the personality disorder assessed? This aspect is not mentioned in the line 116, either. What does “polytherapy” mean in this context, i.e., combinations of moodstabilizers or any drug association, like a moodstabilizer and an add-on agent, for example?

Personality disorders were diagnosed using the DSM-IV-TR or DSM-5 criteria, depending on the time of hospitalization (before or after DSM-5 release). Polytherapy has to be intended as any combination of more than a single psychotropic drug.

Both points have been clarified in the manuscript.

6) The impact of pharmacological agents on metabolic parameters is difficult to assess based on the presented data. Mirtazapine, olanzapine, and tricyclic antidepressants are associated with more frequent weight gain, dyslipidemia, and diabetes than SSRIs, SNRIs, or aripiprazole, but it is not clear how these agents were distributed across genders.

Thanks for the useful observation. No differences between genders were detected in relation to treatment (χ2=20.53, p=0.714), even in relation to the single compounds (p<0.05).

Round 2

Reviewer 1 Report

Comments and Suggestions for Authors

The authors have improved the quality of the article in the results and discussion sections. However, I still believe that the abstract needs to be improved, since it only has 250 words to highlight the results of the article. So far, only 171 words have been used in the abstract.

Author Response

Thank you for the feedback. We have extended the abstract to approximately 250 words, emphasizing the results section.

Reviewer 2 Report

Comments and Suggestions for Authors

The quality of the manuscript significantly improved.

Author Response

Thank you for the feedback.